# *CaGβ* Promotes *CaWRKY40* to Activate Immunity Against *Ralstonia solanacearum* but Disables It from Activating Thermotolerance

**DOI:** 10.3390/plants15010101

**Published:** 2025-12-29

**Authors:** Li He, Meiyun Wan, Xingge Cheng, Xueqiong Chen, Chenfeng Duan, Shuilin He, Yang Wu, Sheng Yang, Ailian Qiu

**Affiliations:** 1College of Agriculture, Fujian Agriculture and Forestry University, Fuzhou 350002, China; heliswjsa@126.com (L.H.); 22401117015@fafu.edu.cn (M.W.); 22401117014@fafu.edu.cn (X.C.); 52405043051@fafu.edu.cn (X.C.); 52305043032@fafu.edu.cn (C.D.); shlhe201304@aliyun.com (S.H.); 2College of Life Sciences, Jinggangshan University, Ji’an 343009, China

**Keywords:** pepper, *Ralstonia solanacearum*, thermotolerance, WRKY, G protein

## Abstract

It has previously been found that *CaWRKY40* is employed by pepper to activate immunity against *Ralstonia solanacearum* and to activate thermotolerance context-specifically, but the underlying mechanisms are not fully understood. Here, CaGβ, a subunit in the heterotrimeric G protein complex that was originally found to probably interact with CaWRKY40, was expressional and functionally characterized; the results showed that *CaGβ* was upregulated by *R. solanacearum* infection; its silencing by virus-induced silencing impaired pepper immunity against *R. solanacearum* infection, accompanied by downregulation of immunity-related marker genes, including *CaPR1*, *CaDEF1*, *CaNPR1*, and *CaPR-STH2*. In addition, CaGβ–CaWRKY40 interaction was confirmed by BiFC and pull-down assay using prokaryotically expressed proteins, and activations of immunity-related *CaPR1*, *CaPR-STH2*, and *CaNPR1* by *CaWRKY40* were all promoted, but the activation of thermotolerance-related *CaHSP24* by *CaWRKY40* blocked *CaGβ* through its interaction with *CaWRKY40*. All these data indicate that immunity against *R. solanacearum* and its antagonism to thermotolerance in pathogen-infected pepper plants are mediated by *CaWRKY40* through physical interaction with *CaGβ.*

## 1. Introduction

In their natural habitats, plants are exposed to attack from pathogens and various abiotic stresses, and have to activate defense responses to survive these stresses [1]. Some of these biotic and abiotic stresses are closely related; for example, attack of pathogen infection and high-temperature stress are two important stresses exposed by plants in tropical and subtropical regions; they generally occur individually and sequentially or in combinations and closely related with each other, with pathogen infection and plant immunity being significantly affected by high temperature [2,3]. These pathogens and abiotic stresses not only affect plant metabolic activities but also impact plant growth and development, potentially leading to plant death [4]. *R. solanacearum* constitutes a globally devastating soil-borne bacterial disease with a broad host range, causing significant losses in crop production and yield [5]. Through prolonged interaction with *R. solanacearum*, plants have evolved a series of complex defense mechanisms against bacterial invasion [6,7]. It has been well established that plant defense responses to biotic and abiotic stresses are largely regulated at the transcriptional level with the actions of various transcription factors, which integrate upstream signaling initiated upon stress perception [8]. Plants possess a remarkable capacity to perceive diverse environmental signals and respond to external stresses and pressures [9]. However, how specific defense response to a given stress and how defense responses to various closely related stresses are coordinately activated by these TFs remain elusive.

WRKY proteins characterized by the conserved WRKY domain constitute one of the largest transcription factor families; the members of this family have been implicated in plant thermotolerance and plant immunity as positive or negative regulators typically by binding a cis-element termed as W-box [10,11]. Plants adapt to complex external environments through intricate regulatory networks. Multiple abiotic factors can induce WRKY transcription factors to regulate associated genes, thereby enhancing plant tolerance to both biotic and abiotic stresses [12]. Analyzing the working mode of WRKY transcription factors is crucial to understanding the stress resistance mechanism of plants. Importantly, some of these WRKY TFs, such as WRKY17, WRKY28, and WRKY40 in pepper, have been found to participate in both thermotolerance and immunity against *R. solanacearum* in pepper with WRKY40 acting as a crucial hub in the signaling network, indicating that the two processes are coordinately activated by WRKY40 [13]. Current studies have shown that WRKY transcription factors are regulated by a variety of interacting proteins, including VQ proteins, transcription factors, CDPK, etc., which are essential for WRKY transcription factor targeting and transcriptional regulation [14]. However, as immunity and thermotolerance antagonize each other, how WRKY40 specifically activates immunity and represses thermotolerance upon pathogen infection remains largely unknown, despite the fact that WRKY40 is modulated by regulatory proteins such as ASR1 to activate immunity specifically upon pathogen infection in the absence of high-temperature stress [15,16].

The heterotrimeric G protein complex comprising Gα, Gβ, and Gγ constitutes one of the most important components of the cell signaling cascade, in which a GDP-bound Gα associates with a Gβγ dimer and G protein coupled receptors (GPCRs) in its inactive state; when activated through ligand perception, the GPCR promotes GDP release and binding of GTP by Gα, activating the G proteins and promoting interaction with downstream effectors, and thus transducing upstream signaling from a diverse range of extracellular stimuli and playing an important role in modulation of various biological processes, including growth, development, and response to biotic and abiotic stresses [17,18]. In addition to the role of Gβ as a Gβγ dimer to keep the heterotrimeric protein complex in an inactive state, Gβ subunit has been found to interact with the CLV-like receptor and appears to participate in modulating multiple biological processes, including development of shoot meristem, floral organ and fruit, salt stress, and nitrogen responses as well as immunity [19,20,21,22,23]. The Gβ subunit positively regulates the ABA-mediated signaling pathway to resist drought stress. In *Nicotiana benthamiana* and *Pisum sativum*, the Gβ subunit exerts a positive regulatory effect on heat stress, with overexpression of Gβ markedly enhancing plant tolerance to thermal stress [24]. Moreover, the Gβ subunit exerts a distinct regulatory function from the Gα subunit during salt stress [25]. However, the direct target of Gβ in modulating these processes remains unexplored so far.

As an important vegetable, pepper has been receiving widespread attention for its disease resistance and thermotolerance [13,26]. In the present study, the data indicate that a Gβ in pepper, acts positively in pepper immunity against *R. solanacearum* by interacting and modulating *CaWRKY40* in promoting its immunity-related target genes but repressing its thermotolerance-related target gene, indicating that *CaGβ* promotes *CaWRKY40* to activate immunity against *R. solanacearum* antagonizing over thermotolerance. Our findings construct a molecular link between Gβ and WRKY transcription factors, providing new insights into pepper immunity.

## 2. Results

### 2.1. CaGβ Exhibit High Sequence Similarity to Its Orthologs in Solanaceas

It was previously found that CaWRKY40 acts positively not only in pepper immune response to *R. solanacearum* but also in thermotolerance; to further elucidate the mechanism underlying the trade-off between immunity and thermotolerance mediated by CaWRKY40, its interacting proteins have been determined by pull-down combined mass spectrometry; a putative Gβ, termed as CaGβ, was found to potentially interact with CaWRKY40. By multiple sequence alignment and phylogenetic tree analysis (Figure 1), CaGβ exhibited high sequence similarity to its orthologs in other solanaceas, such as SdGβ (>XP_055825703.1 *Solanum dulcamara*), SsGβ (>XP_049393750.1, *Solanum stenotomum*), SlGβ (>NP_001233881.2, *Solanum lycopersicum*), LfGβ (>XP_059303414.1, *Lycium ferocissimum*), NpGβ (>CAA96528.1, *Nicotiana plumbaginifolia*), AaGβ (>KAJ8536054.1, *Anisodus acutangulus*), NsGβ (>XP_009776114.1, Nicotiana sylvestris), NtGβ (>XP_009611065.1, *Nicotiana tomentosiformis*), and SvGβ (>XP_049365623.1, *Solanum verrucosum*). So far, no Gβ has been found to be functionally related to a transcription factor.

### 2.2. CaGβ Was Upregulated by R. solanacearum Infection in Pepper Plants

To study the possible role of *CaGβ* in pepper immunity against *R. solanacearum* infection, we first checked the cis-elements within the promoter region of *CaGβ* and found that cis-elements, including that responsible for MYB, Myc binding, and CAT-box as well as CGTCA-box, which are closely related to SA signaling that is frequently related to plant immune response, were found to be present in the promoter of *CaGβ* (Figure 2A). Consistently, the data from RT-qPCR demonstrated that in pepper inbred line HN42 *CaGβ* was upregulated at transcriptional level in *R. solanacearum* inoculation (Figure 2B). These data imply that *CaGβ* might be involved in pepper immunity against *R. solanacearum* infection.

### 2.3. CaGβ Locates in Plasma Membrane, Cytoplasm and Also in the Nuclei

For a given protein, its subcellular location is closely related to its function, to further assay the possible role of CaGβ in pepper immunity, its subcellular location was assayed by transiently overexpressing CaGβ-GFP in epidermal cells of *Nicotiana benthamiana* leaves though the Agrobacterium infiltration-based method, and it was found that the fused protein occurred not only in plasma membrane, cytoplasm but also in the nuclei (Figure 3), consistent with the possibility that CaGβ might interact with CaWRKY40.

### 2.4. CaGβ Silencing by VIGS Significantly Increased Susceptibility of Pepper Plants to R. solanacearum Infection

To assay the possible role of CaGβ in pepper immunity, we silenced CaGβ using virus-induced gene silencing in pepper plants (Figure 4A). It was found that CaGβ was upregulated significantly in TRV::*00* pepper plants, but this upregulation was significantly repressed in TRV::*CaGβ* (Figure 4B), indicating that CaGβ in TRV::*CaGβ* plants was successfully silenced. It was further found that the TRV::*CaGβ* plants exhibited significantly increased susceptibility to *R. solanacearum* infection (Figure 4C); consistently, increased bacterial growth (Figure 4E) and higher dynamic disease index (Figure 4D) were found in *R. solanacearum*-inoculated TRV::*CaGβ* plants compared to the TRV::*00* pepper plants. These results indicated that decreasing the transcription level of CaGB significantly increased the sensitivity of pepper inbred line HN42 to bacterial wilt.

In addition, immunity-related marker genes, including *CaPR1*, *CaNPR1*, *CaDEF1*, and *CaPR-STH2*, which act positively in pepper immunity against *R. solanacearum*, were upregulated by *R. solanacearum* infection, and these upregulations were all repressed by CaGβ silencing (Figure 4F). All these data indicate that CaGβ acts positively in pepper immunity against *R. solanacearum*.

### 2.5. CaGβ Interacted with CaWRKY40 and Promoted CaWRKY40 in Activating Immunity by Blocking Its Activating Thermotolerance-Related CaHSP24

As CaGβ was originally found in the interacting proteins of CaWRKY40, to confirm this interaction we first performed a BiFC assay and found that CaGβ interacted with CaWRKY40 in the nuclei in vivo (Figure 5A). To further confirm CaGβ–CaWRKY40, we performed a pull-down assay using prokaryotically expressed CaGβ-GST and CaWRKY40-6×His; the result also showed that these proteins interacted with each other (Figure 5B). Through AlphaFold3, CaWRKY40 was predicted to interact with CaGβ; they directly exist in four possible direct combination areas (Figure 5C). All these data indicate that CaGβ interacts with CaWRKY40 in the nuclei.

We further assayed the possible effect of CaGβ–CaWRKY40 interaction on *CaWRKY40* transcriptionally regulating immunity and thermotolerance, and found that the transient overexpression of *CaWRKY40* alone upregulated both the tested immunity marker genes and thermotolerance-related *CaHSP24*, while *CaGβ* transient overexpression alone only upregulated immunity-related marker genes but did not affect the transcription of *CaHSP24*; importantly, it was found that the co-transient overexpression of *CaWRKY40* and *CaGβ* activate a higher transcriptional level of immunity-related marker genes than their individual transient overexpression, but *CaGβ* transient overexpression repressed the upregulation of *CaHSP24* by *CaWRKY40* (Figure 5D–F). Through ChIP-qCPR experiments, we uncovered that the silencing of *CaGβ* leads to a significant diminution in the interaction of CaWRKY40 with the promoters of downstream target genes *CaPR1*, *CaNPR1*, and *CaPR-STH2*, whereas the binding of CaWRKY40 to the promoter of *CaHSP24* is significantly enhanced (Figure 5G). This observation suggests that *CaGβ* has an impact on the binding of CaWRKY40 to the promoters of the downstream target genes.

## 3. Discussion

The role of heterotrimeric G protein complexes in plant immune system has been widely considered [27,28]. As its crucial role in signaling implicated in plant growth, development, and defense response to stresses, heterotrimeric G protein complex has been intensively studied, and Gβ has been found to act invariably as a component in this complex as a Gβ-γ dimer to associate with GDP-bound Gα; however, whether and how Gβ acts alone remain unexplored.

The role of Gβ in pepper immune process has not been fully studied. The data in the present study found for the first time that *CaGβ* is upregulated in pepper plants by attack of *R. solanacearum* and acts positively in pepper immunity against *R. solanacearum*, supporting the notion of previous studies that a gene transcriptionally modified by a given stress generally plays a role in the defense response to this stress. The function of heterologous G protein trimer as a core signaling hub in the plant immune system has been fully revealed. GB, as a core member, is widely involved in heterologous G protein trimer-mediated immune signal transduction [19,29,30]. In addition, the role of *CaGβ* in pepper immunity against *R. solanacearum* is closely related to immunity-related marker genes, including SA signaling responsive *CaPR1* and *CaNPR1* as well as JA signaling-related *CaDEF1*, which have been found to act positively in pepper immunity against *R. solanacearum*, indicating that *CaGβ* plays a role in pepper immunity against *R. solanacearum*. These results were similar to those of CaWRKY40, CabZIP63, and other positive regulatory proteins previously found in pepper, indicating that *CaGβ* may be involved in the immune signaling pathway established by these proteins [31]. Our follow-up studies also revealed that CaGB plays a role in pepper defense against *R. solanacearum* by interacting with CaWRKY40 in the nucleus to promote its transcriptional regulation of *CaPR1*, *CaNPR1*, and *CaPR-STH2*. These results suggest that Gβ may be involved in transcriptional regulation independently as a regulatory protein of transcription factors.

Our previous studies have found that pepper employs shared players such as *CaWRKY40*, *CabZIP63*, and *CaCDPK15* in immunity and thermotolerance despite the fact that they antagonize each other [13,31,32,33,34,35,36]. To activate accurately a context-specific defense response, the plant implements rapid and accurate post-translational regulation of these common components through protein–protein interaction [32,35]. In the present study, we found for the first time that *CaGβ* acts as a transcriptional co-activator of *CaWRKY40* in activating immunity while disabling it from activating thermotolerance in pepper plants challenged with *R. solanacearum*, similar to the results of a previous study showing that *CaASR1* promotes salicylic acid, but represses jasmonic acid-dependent signaling to enhance the resistance of *Capsicum annuum* to bacterial wilt by modulating *CabZIP63* [16]. During pepper growth, high temperature, high humidity, and *R. solanacearum* infection often occur simultaneously or alternately. It is very important for pepper and other Solanaceae plants to activate the corresponding defense response to different stresses [32]. Interaction with different regulatory proteins is likely to be an important way of ensuring that transcription factors with multiple functions perform their functions accurately. It remains to be determined whether CaGβ functions via the Gβγ dimer or synergistic interactions with other G protein subunits (e.g., Gα, Gγ). In addition, investigating whether CaGβ indirectly modulates immunity and thermotolerance by regulating other transcription factors could help clarify whether its regulation of CaWRKY40 is a direct, specific interaction, or part of a more complex regulatory network.

We uncover that *CaGβ* acts positively in pepper immunity against *R. solanacearum* infection by promoting *CaWRKY40* via physical interaction in activating SA- and JA-dependent immunity-related genes while disabling it from activating thermotolerance. This further enriches our understanding of CaWRKY40-mediated transcriptional regulation in pepper thermotolerance and bacterial wilt resistance immune responses.

## 4. Materials and Methods

### 4.1. Plant Materials and Growth Conditions

Seeds of pepper inbred line HN42 and *Nicotiana benthamiana* were sown in a soil mix [peat moss: perlite, 3:1 (*v*/*v*)]. When the seeds germinated and the cotyledon of each plant fully unfolded, each seedling was planted in a small flowerpot and placed under conditions of 25 °C, 60–70 µmol photons m^−2^ s^−1^, 70% relative humidity, and 16 h of light/8 h of dark light.

### 4.2. Pathogen and Pathogen Inoculation

A highly virulent *R.solaanacearum* strain GMI1000 was used; the bacterial suspension used for inoculation was diluted to 10^8^ colony-forming units mL^−1^ (OD_600_ = 0.6–0.8). Root irrigation with the *R. solanacearum* suspension followed the method described previously by Cai et al. [32].

### 4.3. Vector Construction

The vector construction in the present study was also performed using Gateway technology (Thermo Fisher Scientific, Waltham, MA, USA). The full-length open reading frame (ORF) of CaGβ or CaWRKY40 (with or without termination codon) was first cloned into the entry vector pDONR207 by BP reaction and then transferred into distinction vectors, including pEarlyGate101, pSPYNE, pSPYCE, pDEST17, or pDEST15 by LR reaction. To construct the vector for CaGβ silencing, a specific cDNA fragment of around 300 bp in length was amplified by PCR with a specific primer pair (Appendix A), which was further transferred first into entry vector pDONR207 and then into tobacco rattle virus TRV2 vector PYL279 following the method described above.

### 4.4. Silencing of CaGβ by Virus-Induced Gene Silencing (VIGS) in Pepper Plants

To silence CaGβ in pepper plants using the VIGS approach, Agrobacterium strain GV3101 cells containing PYL192 and PYL279-CaGβ were suspended in induction medium following the method used and described previously [16,33]; briefly, GV3101 cells containing PYL192 and cells containing PYL279-CaGβ were mixed at a 1:1 ratio and were co-infiltrated into cotyledons of 2-week-old pepper plants; then the agrobacterium infiltrated plants were incubated in darkness at 16 °C, 70% humidity for 56 h, and then placed in a growth chamber under the conditions of 25 °C, 60–70 µmol photons m^−2^ s^−1^, 70% relative humidity, and a 16 h light/8 h dark photoperiod. Around 4 weeks later, the efficiency of gene silencing was monitored by RT-qPCR using a specific primer pair.

### 4.5. Agrobacterium-Mediated Transient Overexpression and Subcellular Localization as Well as Bimolecular Fluorescent Complimentary (BiFC) Assay

For the subcellular localization assay, GV3101 cells containing 35S::*CaGβ-YFP* (using 35S::*YFP* as control) were suspended using the induction solution (200 mM acetosyringone, 10 mM MES, 10 mM MgCl_2_, pH5.4) and diluted to OD_600_ = 0.6–0.8 and then incubated at room temperature at low speed for 1–3 h. Using disposable sterile syringes without needles, the suspension was injected into *N. benthamiana* leaves and incubated under the specified conditions for approximately 48 h. Small sections of intact *N. benthamiana* leaves were cut and placed back-side up on microscope slides for mounting and examined for fluorescence signals under a Leica laser confocal microscope. Images were captured for documentation.

For the BiFC assay, GV3101 cells containing CaGβ-YFP^c^ and CaWRKY40-YFP^n^ (using CaGβ-YFP^c^ + YFP^n^ and CaWRKY40-YFP^n^ + YFP^c^ as control) were diluted to OD_600_ = 0.6–0.8 using the induction solution (200 mM acetosyringone, 10 mM MES, 10 mM MgCl_2_, pH5.4) and were mixed in a 1:1 ratio; the mixed GV3101 cell solution was placed on a shaker at 100 rpm at 25 °C for 2–3 h, and then the cell solutions were infiltrated into leaves of pepper or *N. benthamiana* leaves using syringes without needles; the leaves were harvested at 48 h post-inoculation, and the YFP signal was observed under a LEICA TCS confocal laser microscope (Leica Microsystems GmbH, Wetzlar, Germany).

### 4.6. Determination of CFU of Ralstonia solanacearum

Inoculate pepper or tobacco leaves using the leaf-puncture method with *Ralstonia solanacearum*. After 24 h and 48 h of cultivation, rinse leaves twice with distilled water, blot dry with filter paper, and sample using a 0.6 mm punch (avoiding *R.solaanacearum* injection sites) in a sterile laminar flow hood. Place 6 small circular pieces per leaf into 2 mL centrifuge tubes containing 500 μL sterile water, with 4 biological replicates per group; crush the samples using a sterile grinding rod, add 500 μL sterile water, and perform sequential serial dilutions on the samples. Pipette 400 μL aliquots of the 10^−3^, 10^−4^, and 10^−5^ dilutions onto TM agar plates. Label the plates, air-dry the dilutions, invert the plates, and incubate at 28 °C for 2 days. Count the colonies on the plates.

### 4.7. Pull Down Assay and Western Blotting

The CaGβ–CaWRKY40 interaction was assayed in vitro by pull-down assay following the method described previously [13,32]. CaGβ-GST and CaWRKY40-6×His proteins, which were prokaryotically expressed in an isolated from *E. coli* strain BL21, were mixed with BeaverBeads IDA-Nickel (Beaver Biosciences, Suzhou, China) and incubated at 4 °C on a shaking incubator for 2–3 h. The proteins were removed, and the magnetic beads were washed 3–4 times with washing buffer to remove surplus proteins. Subsequently, the proteins bound to the magnetic beads were eluted using elution buffer. Add 5× SDS loading buffer to the eluted sample and denature at 95 °C for 10 min. Separate proteins on an SDS-PAGE gel, then transfer the proteins to a PVDF membrane. Analyze the presence of interacting proteins by Western blotting, following the method of Cheng et al. [35] using GST and His antibodies (Sigma-Aldrich, St. Louis, MO, USA).

### 4.8. RNA Extraction and RT-qPCR Assay

Total RNA extraction from pepper and *N. benthamiana* plants was performed using TRIpure reagent (TransGen, Biotech, Beijing, China) following the method used previously [13]. The extracted RNA concentration was standardized, and cDNA templates were synthesized utilizing reverse transcriptase (Vazyme, Nanjing, China). Relative transcript levels of genes were detected using ChamQ Blue Universal SYBR qPCR Master Mix on a Bio-Rad real-time polymerase chain reaction system (Bio-Rad Laboratories, Hercules, CA, USA) using specific primer pairs. The normalization of transcript levels was performed according to the Livak method [37], and internal reference gene *CaActin* was employed.

## Figures and Tables

**Figure 1 plants-15-00101-f001:**
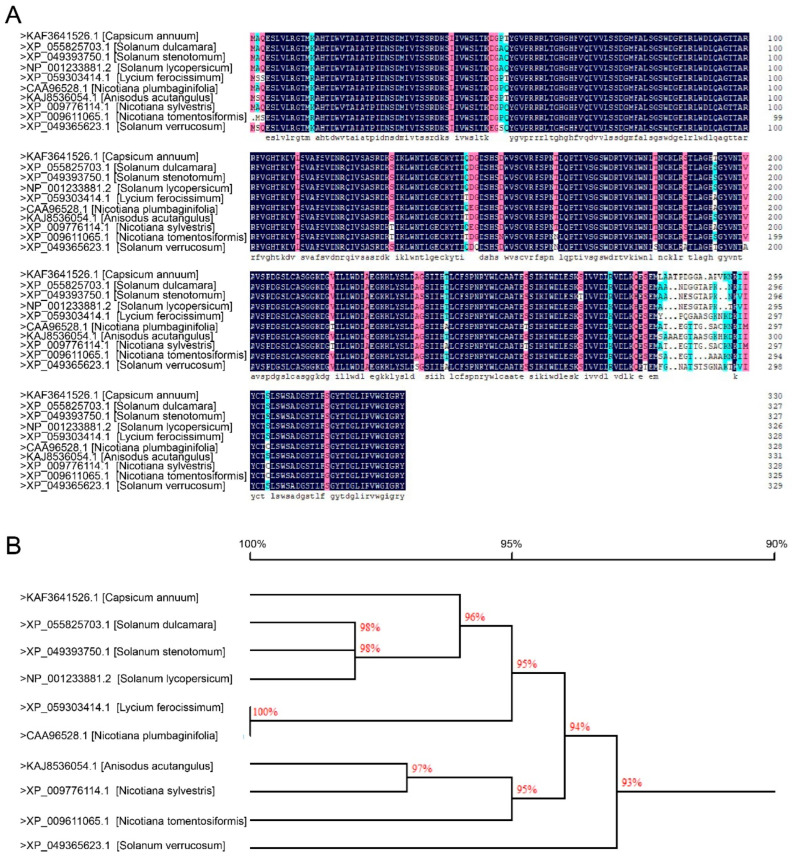
CaGβ exhibits high sequence similarity to its orthologs in other solanaceaous plants. (**A**) Comparison of deduced amino acid sequence of CaGβ with that of its orthologs in other solanaceaous plants. Green shading indicates 50–70% similarity, red shading denotes 75–100 similarity, and black shading represents 100% similarity. Alignment was carried out by DNAMAN8. (**B**) Phylogenetic analysis of CaGβ with its orthologues in other solanaceaous plants.

**Figure 2 plants-15-00101-f002:**
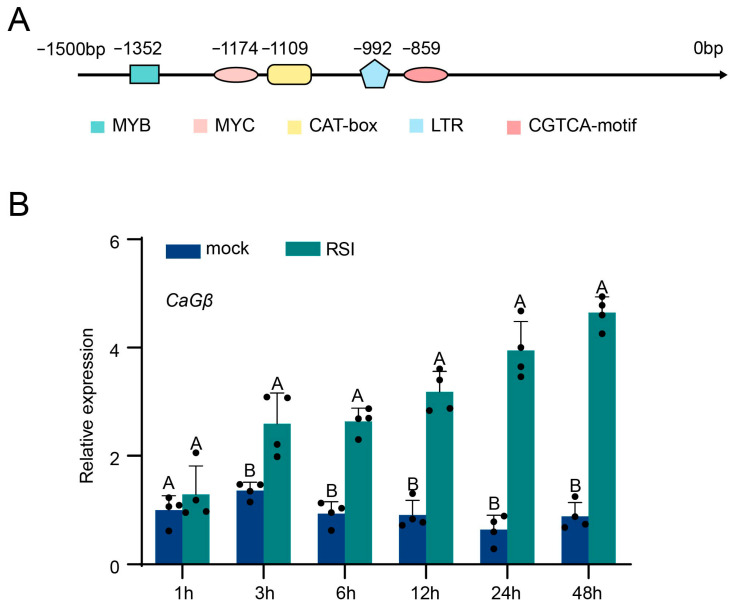
Cis-elements in the promoter region of *CaGβ* and its transcriptional response to *R. solanacearum* inoculation. (**A**) Cis-elements in the promoter region of *CaGβ*, which are determined by searching online (PlantCARE, a database of plant promoters and their cis-acting regulatory elements). (**B**) Relative transcript levels of *CaGβ* in leaves of pepper plants challenged with RSI by RT-qPCR. Data presented are mean standard deviation of four replicates. Different uppercase letters above the bars indicate significant differences among means (*p* < 0.01) by Fisher’s protected least significant difference (LSD) test.

**Figure 3 plants-15-00101-f003:**
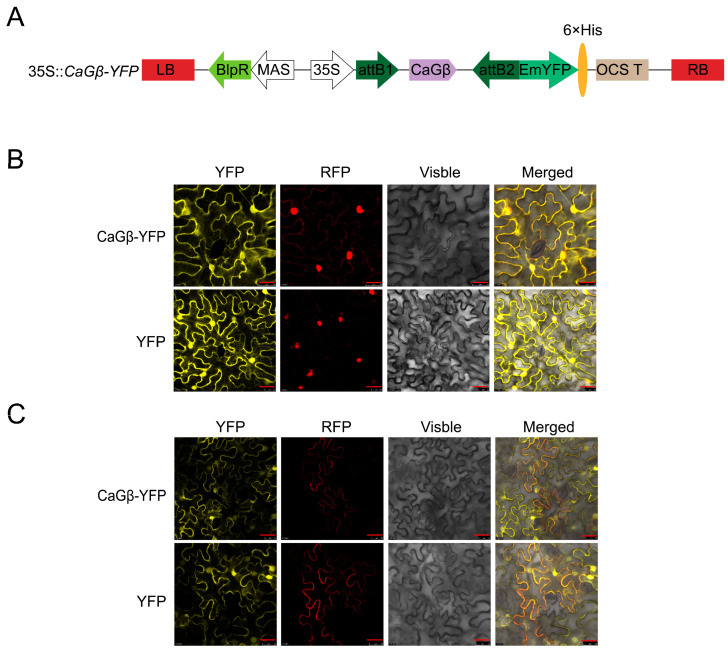
Subcellular localization of CaGβ in epidermal cells in *N. benthamiana* leaves. (**A**) The diagram of 35S::*CaGβ-YFP* fusion expression vector. (**B**,**C**) *N. benthamiana* leaves were infiltrated with GV3101 cells harboring 35S::*CaGβ-YFP* (using 35S::*YFP* as control). Subcellular localization of CaGβ-YFP or GFP was observed with a laser scanning confocal microscope at 48 hpi. The nuclei were displayed by 35S::*H2B-RFP* (**B**), the plasma membranes were displayed by 35S::*CaSRC2-RFP* (**C**), fluorescence image (YFP), bright-field images (visible), and the corresponding overlay images (merged) of representative cells expressing YFP or CaGβ-YFP fusion protein are shown. Bars = 25 μm.

**Figure 4 plants-15-00101-f004:**
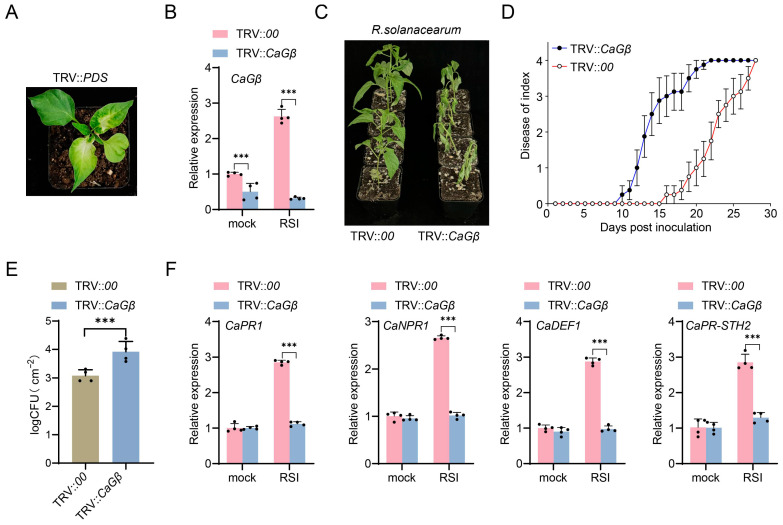
*CaGβ* silencing increased pepper susceptibility to *R. solanacearum* infection. (**A**) Constructing TRV::*CaPDS* as an indicator plant, which exhibits white leaves approximately 25–30 days after treatment. (**B**) CaGβ was successfully silenced in TRV::*CaGβ* pepper plants via virus-induced gene silencing (VIGS) by RT-qPCR at 48 h post-inoculation. (**C**) TRV::*CaGβ* pepper plants exhibited increased susceptibility to *R. solanacearum* inoculation compared to TRV::*00* at 14 days post-inoculation by root irrigation. (**D**) TRV::*CaGβ* pepper plants exhibited higher level of dynamic disease index from 0 to 30 days post-inoculation. (**E**) TRV::*CaGβ* pepper plants supported a high level of *R. solanacearum* growth displayed by cfu (clone-forming units) compared to TRV::*00* pepper plants at 24 h post-inoculation. (**F**) Relative transcript levels of immunity-related genes, including *CaPR1*, *CaNPR1*, *CaDEF1*, and *CaPR-STH2* were significantly downregulated by *CaGβ* silencing at 48 h post-inoculation. In (**B**,**D**–**F**), data presented are means standard deviation of four replicates (**B**,**E**,**F**) or 10 individual plants (**D**). The *** above the bar chart indicates a significant difference between means as determined by the *t*-test (*p* < 0.01).

**Figure 5 plants-15-00101-f005:**
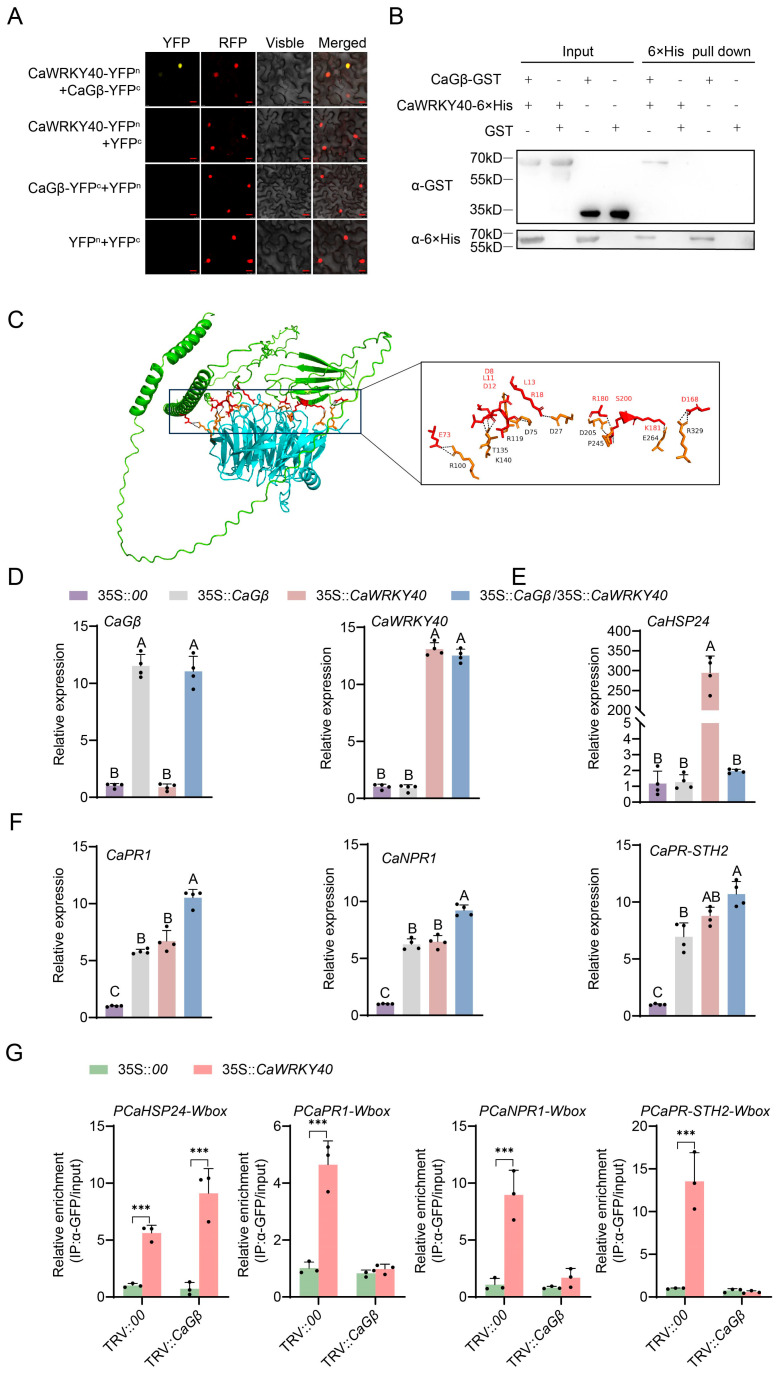
CaGβ–CaWRKY40 interaction in vivo and in vitro and the effect of this interaction on the transcriptional regulation of *CaWRKY40* to different target genes. (**A**) Interaction between CaGβ and CaWRKY40 in epidermal cells in *N. benthamiana* leaves in a BiFC assay. (**B**) Pull-down assay revealing the interaction between CaGβ and CaWRKY40 interaction in vitro. (**C**) Using Pymol software 3.0 for binding site analysis, it was found that both proteins exhibit binding at distinct amino acid sites, with red representing CaWRKY40 and yellow representing CaGβ. (**D**) By RT-qPCR, both *CaGβ* and *CaWRKY40* were successfully transiently overexpressed in pepper leaves via agro-infiltration. (**E**) Thermotolerance-related *CaHSP24* was upregulated by transient overexpression of *CaWRKY40*, but this upregulation was repressed by the co-transient overexpression of *CaGβ*. (**F**) Immunity-related genes, including *CaPR1*, *CaNPR1*, and *CaPR-STH2*, were upregulated by transient overexpression of *CaWRKY40*, and these upregulations were also promoted by co-transient overexpression of *CaGβ*. (**G**) ChIP-qPCR to assess the effect of *CaGβ* silencing on *CaWRKY40* binding to *CaHSP24*, *CaPR1*, *CaNPR1*, and *CaPR-STH2*. In (**C**–**F**), data presented are means standard deviation of four replicates. Different uppercase letters above the bars indicate significant differences among means (*p* < 0.01) by Fisher’s protected LSD test. The *** above the bar chart indicates a significant difference between means as determined by the *t*-test (*p* < 0.01).

## Data Availability

The original contributions presented in this study are included in the article/Appendix A. Further inquiries can be directed to the corresponding author.

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
