# Peer review of "CaGβ* Promotes *CaWRKY40* to Activate Immunity Against *Ralstonia solanacearum* but Disables It from Activating Thermotolerance"

_plants, 2025, doi:10.3390/plants15010101_

Round 1

Reviewer 1 Report

Comments and Suggestions for Authors

Plants frequently face concurrent biotic and abiotic stresses in natural environments, and the trade-off between disease resistance and abiotic stress tolerance has become a key focus in plant stress biology research. This manuscript investigates the regulatory role of CaGβ, a heterotrimeric G protein subunit, in mediating the interaction between CaWRKY40 and the pepper responses to R. solanacearum infection and thermotolerance. The research direction is relevant and addresses an important biological question, with a sound overall experimental design that includes sequence analysis, gene expression profiling, subcellular localization, and protein-protein interaction assays. However, several critical issues and minor deficiencies need to be addressed to enhance the scientific rigor, completeness, and clarity of the study.

Major Issues

  1. It would be beneficial to explore additional potential regulatory scenarios to refine the specificity of CaGβ-CaWRKY40 interaction. Whether CaGβ exerts its effect through the Gβγ dimer or synergistic interactions with other G protein subunits (e.g., Gα, Gγ) remains to be examined. Additionally, investigating whether CaGβ might indirectly modulate immunity and thermotolerance by regulating other transcription factors could help confirm whether its regulation of CaWRKY40 is a direct, specific interaction or part of a more complex regulatory network.

  1. The current study focuses on the single stress of R. solanacearum, which is a valuable starting point. However, in natural growth environments, pathogen infection and high temperature often occur simultaneously or alternately. Exploring the expression pattern of CaGβ and its regulatory effects on CaWRKY40 under the combined stress of "R. solanacearum + high temperature" would enhance the relevance of the findings to real-world scenarios and further elevate the study practical and ecological value. I am curious about the expression and regulatory patterns of CaGβ under combined stress. Please briefly address this in the response or discuss it in the discussion section.

Minor Issue

  1. Spelling error: "previsously" in the abstract should be "previously".

  1. Spelling errors in the Introduction:

"abiotic streses" should be "abiotic stresses",

"defense resposnes" should be "defense responses",

"intergrate" should be "integrate",

"characterzied" should be "characterized".

  1. Spelling errors in the Results:

"othologs" should be "orthologs",

"Comparsion" should be "Comparison",

"Phylogenetic analylsis" should be "Phylogenetic analysis".

  1. Spelling errors in Materials and Methods:

"vector contruction" should be "vector construction",

"contructions" should be "constructions",

"Root irrgation" should be "Root irrigation",

"disolated" should be "isolated",

"streain" should be "strain".

  1. Some figure legends lack clear descriptions of details (e.g., Figure 4E does not specify the gene names corresponding to each subfigure). These labels should be added directly to the figures for clarity.

  1. The reference format is inconsistent: some references lack page numbers (e.g., Gupta et al. 2017).
Comments on the Quality of English Language

The text contains frequent spelling errors in common words, affecting its readability and standardization.

Author Response

Plants frequently face concurrent biotic and abiotic stresses in natural environments, and the trade-off between disease resistance and abiotic stress tolerance has become a key focus in plant stress biology research. This manuscript investigates the regulatory role of CaGβ, a heterotrimeric G protein subunit, in mediating the interaction between CaWRKY40 and the pepper responses to R. solanacearum infection and thermotolerance. The research direction is relevant and addresses an important biological question, with a sound overall experimental design that includes sequence analysis, gene expression profiling, subcellular localization, and protein-protein interaction assays. However, several critical issues and minor deficiencies need to be addressed to enhance the scientific rigor, completeness, and clarity of the study.

Major Issues

  1. It would be beneficial to explore additional potential regulatory scenarios to refine the specificity of CaGβ-CaWRKY40 interaction. Whether CaGβ exerts its effect through the Gβγ dimer or synergistic interactions with other G protein subunits (e.g., Gα, Gγ) remains to be examined. Additionally, investigating whether CaGβ might indirectly modulate immunity and thermotolerance by regulating other transcription factors could help confirm whether its regulation of CaWRKY40 is a direct, specific interaction or part of a more complex regulatory network.

Response: Thank you very much for your good suggestion. We have incorporated this content into the discussion section, and we will further elucidate the mechanisms of action of CaGβ in accordance with your recommendations in the future.

  1. The current study focuses on the single stress of R. solanacearum, which is a valuable starting point. However, in natural growth environments, pathogen infection and high temperature often occur simultaneously or alternately. Exploring the expression pattern of CaGβ and its regulatory effects on CaWRKY40 under the combined stress of "R. solanacearum + high temperature" would enhance the relevance of the findings to real-world scenarios and further elevate the study practical and ecological value. I am curious about the expression and regulatory patterns of CaGβ under combined stress. Please briefly address this in the response or discuss it in the discussion section.

Response: Thank you very much for your good suggestion. Our previous research has revealed significant discrepancies in the response of Solanaceae plants, such as chili peppers, to the combined stress of high temperature and bacterial wilt, compared to their individual stresses. Under normal conditions, the salicylic acid (SA) and jasmonic acid (JA) signaling pathways play a crucial role in disease resistance, whereas, in high temperature and high humidity conditions, the cytokinin signaling pathway takes over these functions. We have also explored the role of CaGβ in the composite stress of high temperature and bacterial wilt infection in chili peppers, finding that it acts as a negative regulator under these conditions, differing from its role in single stress conditions. This makes it challenging to integrate this research into our current manuscript. However, we have furthered our investigation on this topic and plan to publish it in future papers.

Minor Issue

  1. Spelling error: "previsously" in the abstract should be "previously".

Response: Thank you very much for your good suggestion. We have amended the misspelt "previsously" to "previously". please see line 13 in the revised MS.

  1. Spelling errors in the Introduction:

"abiotic streses" should be "abiotic stresses",

"defense resposnes" should be "defense responses",

"intergrate" should be "integrate",

"characterzied" should be "characterized".

Response: Thank you very much for your good suggestion. We have amended the misspelt "abiotic streses" to "abiotic stresses"、"defense resposnes" to "defense responses"、"intergrate" to "integrate" and "characterzied" to "characterized". please see line 31、44、47 and 50 in the revised MS.

  1. Spelling errors in the Results:

"othologs" should be "orthologs",

"Comparsion" should be "Comparison",

"Phylogenetic analylsis" should be "Phylogenetic analysis".

Response: Thank you very much for your good suggestion. We have amended the misspelt "othologs" to "orthologs"、"Comparsion" to "Comparison" and "Phylogenetic analylsis" to "Phylogenetic analysis. please see line 106、116 and 119 in the revised MS.

  1. Spelling errors in Materials and Methods:

"vector contruction" should be "vector construction",

"contructions" should be "constructions",

"Root irrgation" should be "Root irrigation",

"disolated" should be "isolated",

"streain" should be "strain".

Response: Thank you very much for your good suggestion. We have amended the misspelt "vector contruction" to "vector construction"、"Root irrgation" to "Root irrigation"、"disolated" to "isolated" and "streain" to "strain". please see line 298-299、302-303 and 357 in the revised MS.

  1. Some figure legends lack clear descriptions of details (e.g., Figure 4E does not specify the gene names corresponding to each subfigure). These labels should be added directly to the figures for clarity.

Response: Thank you very much for your good suggestion. Each diagram clearly labels the genes.

  1. The reference format is inconsistent: some references lack page numbers (e.g., Gupta et al. 2017).

Response: Thank you very much for your good suggestion. We have supplemented the missing page numbers in the references. Please review the revised section of the bibliography.

Reviewer 2 Report

Comments and Suggestions for Authors

The manuscript entitled “CaGβ promotes CaWRKY40 to activate immunity against R. solanacearum but disables it from activating thermotolerance” is interesting and presents valuable findings. However, several issues should be carefully addressed before it can be considered for publication.

Major issues

  1. The manuscript claims the “first evidence” that CaGβ modulates CaWRKY40 in immunity and thermotolerance. However: Several prior publications already show multiple CaWRKY40 interactors in this pathway. The conceptual novelty beyond previously known regulators (CaASR1, CaSTH2, CabZIP63, CaCDPK29, NAC2c) is not clearly established.
  2. The authors conclude CaGβ “promotes immunity but disables thermotolerance” via CaWRKY40. However:No ChIP-qPCR or promoter binding assays were performed to show direct regulatory impact on CaWRKY40 DNA binding. No domain-mapping was done to identify interaction regions.
  3. The VIGS construct uses a ~300 bp fragment, but:No off-target analysis is shown.No phenotypic controls (TRV::PDS positive control) are provided.
  4. CaGβ is shown in membrane, cytoplasm, and nucleus, which is unusual for Gβ proteins.
    But: No co-localization markers for plasma membrane vs cytoplasm were included. Quantification of fluorescence localization is missing.
  5. The pull-down assay uses recombinant proteins, but: No negative controls (e.g., GST alone) are shown. Interaction strength or binding affinity is not measured. BiFC lacks nuclear marker image merging for precise localization.
  6. The authors propose CaGβ “co-activates” CaWRKY40, yet:Transient assays do not use reporter constructs (e.g., W-box::LUC). Effects on CaWRKY40 protein level, stability, or phosphorylation are not examined.
  7. Error bars unclear (SD vs SE not consistently explained). Sample size for some assays (n=4) too small.No description of biological vs technical replicates.
  8. The study focuses on CaHSP24, but:No heat treatment experiments were conducted.It is unclear whether “thermotolerance repression” actually occurs physiologically.
  9. Numerous grammatical errors reduce clarity and scientific professionalism.
    Examples include: “upegulation,” “repressing thermotolernace,” “exhibit high sequence similarity,” “silencing by virus induced silencing impaired pepper immunity…”Run-on sentences in the Introduction affect readability.

Minor concerns

  1. CaGβ, CaWRKY40, CaPR1, etc. should follow uniform formatting.
  2. PlantCARE analysis should include:Length of promoter used-Statistical relevance (enrichment analysis)
  3. Legends should specify:Time points; Infiltration OD; Conditions for imaging
  4. Light intensity units incorrect (“mmol photons m−2 s−1”—should be µmol).
  5. No reference for disease index scoring criteria.
  6. Pull-down Buffer Composition Not Provided; This limits reproducibility.
  7. Western blot images in figures should include: Marker bands: Full blot panels
  8. Typographical errors throughout the draft “imunity” instead of “immunity”

“G protein conracted” “the two processes are coordiately activated”

Comments on the Quality of English Language

Should be improved

Author Response

The manuscript entitled “CaGβ promotes CaWRKY40 to activate immunity against R. solanacearum but disables it from activating thermotolerance” is interesting and presents valuable findings. However, several issues should be carefully addressed before it can be considered for publication.

Major issues

  1. The manuscript claims the “first evidence” that CaGβ modulates CaWRKY40 in immunity and thermotolerance. However: Several prior publications already show multiple CaWRKY40 interactors in this pathway. The conceptual novelty beyond previously known regulators (CaASR1, CaSTH2, CabZIP63, CaCDPK29, NAC2c) is not clearly established.

Response: Thank you very much for your good suggestion. We had not previously identified the G-protein complex as being involved in the immune response process mediated by CaWRKY40. We have revised our statement to make it more accurate, please see line 276 in the revised MS.

  1. The authors conclude CaGβ “promotes immunity but disables thermotolerance” via CaWRKY40. However:No ChIP-qPCR or promoter binding assays were performed to show direct regulatory impact on CaWRKY40 DNA binding. No domain-mapping was done to identify interaction regions.

Response: Thank you very much for your good suggestion. We have supplemented the ChIP-qPCR experiments, which convincingly demonstrate that silencing CaGB leads to a significant inhibition of CaWRKY40's binding to the PR gene promoter's W-box, while simultaneously promoting CaWRKY40's binding to the CaHSP24 promoter. This experiment robustly substantiates the role of CaGB in modulating the target specificity of CaWRKY40, thereby affecting its function in activating immunity and suppressing the thermotolerance process, please see line 210-line 215 and fig.5 in the revised MS.

We also employed AlphaFold3 to analyze the interaction between CaWRKY40 and CaGB, which revealed multiple high-confidence interaction sites. Further validation will be conducted in subsequent studies, please see line 199-line 200 and fig.5 in the revised MS.

  1. The VIGS construct uses a ~300 bp fragment, but: No off-target analysis is shown.No phenotypic controls (TRV::PDS positive control) are provided.

Response: Thank you very much for your good suggestion. We did not identify any homologous genes in the pepper genome that are similar to the CaGβ sequence, thus ensuring the specificity of the designed VIGS target segment. We have supplemented the TRV::PDS diagram. please see line 174 in the revised MS.

  1. CaGβ is shown in membrane, cytoplasm, and nucleus, which is unusual for Gβ proteins.
    But: No co-localization markers for plasma membrane vs cytoplasm were included. Quantification of fluorescence localization is missing.

Response: Thank you very much for your good suggestion. We have added membrane markers (CaSRC2-RFP) and demonstrated that there is overlap in the fluorescence between YFP and RFP markers. This result can serve as evidence that CaGβ is also distributed in the plasma membrane, please see line 153-line 154 and fig.3 in the revised MS.

  1. The pull-down assay uses recombinant proteins, but: No negative controls (e.g., GST alone) are shown. Interaction strength or binding affinity is not measured. BiFC lacks nuclear marker image merging for precise localization.

Response: Thank you very much for your good suggestion. We have incorporated negative controls into the pull-down experiments and conducted bimolecular fluorescence complementation (BiFC) assays using nuclear markers, please see line fig.5 A and B in the revised MS.

  1. The authors propose CaGβ “co-activates” CaWRKY40, yet:Transient assays do not use reporter constructs (e.g., W-box::LUC). Effects on CaWRKY40 protein level, stability, or phosphorylation are not examined.

Response: Thank you very much for your good suggestion. In this manuscript, quantitative real-time PCR (qRTPCR) was employed to substantiate that the co-transient overexpression of CaWRKY40 and CaGB leads to a pronounced enhancement in the expression of CaPR1, CaNPR1, and CaPR-STH2, compared to their individual overexpression, suggesting the potential for a synergistic activation mechanism among these genes. However, as you have pointed out, we lack sufficient evidence to definitively demonstrate that CaGB activates CaWRKY40. Therefore, we have revised the relevant sections to reflect this nuanced understanding, please see line line 281 to line 283 in the revised MS.

  1. Error bars unclear (SD vs SE not consistently explained). Sample size for some assays (n=4) too small.No description of biological vs technical replicates.

Response: Thank you very much for your good suggestion. The error bars in the text are all standard deviations; the standard errors have been replaced with standard deviations. please see line 141、195 and 240 in the revised MS.

  1. The study focuses on CaHSP24, but:No heat treatment experiments were conducted.It is unclear whether “thermotolerance repression” actually occurs physiologically.

Response: Thank you very much for your good suggestion. We have previously demonstrated through extensive studies that CaWRKY40 is a positive regulator of heat tolerance in pepper, and CaHSP24 is a downstream target gene directly transcribed by CaWRKY40 during heat tolerance. Therefore, we utilized CaGB to infer the role of CaGB in heat tolerance by examining the effect of CaWRKY40 on the transcriptional activation of CaHSP24.

1.Xingge, Cheng,Meiyun, Wan,Yuqiu, Song et al. CaSTH2 disables CaWRKY40 from activating pepper thermotolerance and immunity against Ralstonia solanacearum via physical interaction.[J] .Hortic Res, 2024, 11: uhae066.

2.Weiwei, Cai,Sheng, Yang,Ruijie, Wu et al. CaSWC4 regulates the immunity-thermotolerance tradeoff by recruiting CabZIP63/CaWRKY40 to target genes and activating chromatin in pepper.[J] .PLoS Genet, 2022, 18: e1010023.

3.Jinfeng, Huang,Lei, Shen,Sheng, Yang et al. CaASR1 promotes salicylic acid- but represses jasmonic acid-dependent signaling to enhance the resistance of Capsicum annuum to bacterial wilt by modulating CabZIP63.[J] .J Exp Bot, 2020, 71: 6538-6554.

4.Sheng, Yang,Yuanyuan, Shi,Longyun, Zou et al. Pepper CaMLO6 Negatively Regulates Ralstonia solanacearum Resistance and Positively Regulates High Temperature and High 5.Humidity Responses.[J] .Plant Cell Physiol, 2020, 61: 1223-1238.

Deyi, Guan,Feng, Yang,Xiaoqin, Xia et al. CaHSL1 Acts as a Positive Regulator of Pepper Thermotolerance Under High Humidity and Is Transcriptionally Modulated by CaWRKY40.[J] .Front Plant Sci, 2018, 9: 1802.

  1. Lei, Shen,Zhiqin, Liu,Sheng, Yang et al. Pepper CabZIP63 acts as a positive regulator during Ralstonia solanacearum or high temperature-high humidity challenge in a positive feedback loop with CaWRKY40.[J] .J Exp Bot, 2016, 67: 2439-51.
  2. Feng-Feng, Dang,Yu-Na, Wang,Lu, Yu et al. CaWRKY40, a WRKY protein of pepper, plays an important role in the regulation of tolerance to heat stress and resistance to Ralstonia solanacearum infection.[J] .Plant Cell Environ, 2012, 36: 757-74.

  1. Numerous grammatical errors reduce clarity and scientific professionalism.
    Examples include: “upegulation,” “repressing thermotolernace,” “exhibit high sequence similarity,” “silencing by virus induced silencing impaired pepper immunity…”Run-on sentences in the Introduction affect readability.

 Response: Thank you very much for your good suggestion. We meticulously examined and corrected spelling and grammatical errors in this manuscript.

Minor concerns

  1. CaGβ, CaWRKY40, CaPR1, etc. should follow uniform formatting.

Response: Thank you very much for your good suggestion. To distinguish between proteins and genes, we employ italics for genes and roman type for proteins in our writing.

  1. PlantCARE analysis should include:Length of promoter used-Statistical relevance (enrichment analysis)

Response: Thank you very much for your good suggestion. We provide the full-length sequence of the promoter in the appendix. please see the revised MS.

  1. Legends should specify:Time points; Infiltration OD; Conditions for imaging.

Response: Thank you very much for your good suggestion. Detailed descriptions are provided in the figure captions and research methodology. please see line 158、345 and 350 in the revised MS.

  1. Light intensity units incorrect (“mmol photons m−2 s−1”—should be µmol).

Response: Thank you very much for your good suggestion. We have changed the unit of luminous intensity to µmol. please see line 302 and 329 in the revised MS.

  1. No reference for disease index scoring criteria.

Response: Thank you very much for your good suggestion. We Conduct an assessment of the pathogenicity index of Ralstonia solanacearum in accordance with the industry standards promulgated by the Ministry of Agriculture of the People's Republic of China.

  1. Pull-down Buffer Composition Not Provided; This limits reproducibility.

Response: Thank you very much for your good suggestion. The Pull-down buffer we use is prepared according to the formulation specified in the in the instructions for the BeaverBeads IDA-Nickel.

  1. Western blot images in figures should include: Marker bands: Full blot panels.

Response: Thank you very much for your good suggestion. We have annotated the size of the proteins in the images and provided the full, overlayed marker original images for your reference, please see Fig.5B in the revised MS.

  1. Typographical errors throughout the draft “imunity” instead of “immunity”

“G protein conracted” “the two processes are coordiately activated”

Response: Thank you very much for your good suggestion. We have rectified the typographical errors present in the manuscript.

Round 2

Reviewer 2 Report

Comments and Suggestions for Authors

Although the scientific content has improved, the English language quality still requires substantial revision. The manuscript contains numerous grammatical errors, awkward sentence constructions, and inconsistencies that affect readability and clarity. A thorough language editing by a native English speaker or a professional editing service is strongly recommended before the next revision.

Comments on the Quality of English Language

Should be improved